# Order–disorder and ionic conductivity in calcium nitride-hydride

G. J. Irvine [1] ✉, Ronald I. Smith [2], M. O. Jones[1,2] & J. T. S. Irvine[1,2] ✉

Recently nitrogen-hydrogen compounds have successfully been applied as co-catalysts for mild conditions ammonia synthesis. $Ca_2NH$ was shown to act as a $H_2$ sink during reaction, with H atoms from its lattice being incorporated into the $NH_3(g)$ product. Thus the ionic transport and diffusion properties of the N–H co-catalyst are fundamentally important to understanding and developing such syntheses. Here we show hydride ion conduction in these materials. Two distinct calcium nitride-hydride $Ca_2NH$ phases, prepared via different synthetic paths are found to show dramatically different properties. One phase ($\beta$) shows fast hydride ionic conduction properties (0.08 S/cm at 600 °C), on a par with the best binary ionic hydrides and 10 times higher than $CaH_2$, whilst the other ($\alpha$) is 100 times less conductive. An in situ combined analysis techniques reveals that the effective β-phase conducts ions via a vacancy-mediated phenomenon in which the charge carrier concentration is dependent on the ion concentration in the secondary site and by extension the vacancy concentration in the main site.

Recently, nitrogen–hydrogen (N–H) compounds have attracted interest as potential co-catalysts in mild conditions ammonia synthesis (MCAS: 0.1 MPa, <300 °C)[1–8]. The long standing issue for MCAS to overcome relates to a scaling relation, due to the saturation of the transition metal (Ru, Co, Ni, and Fe) with adatoms of $H_2$, $N_2$, and $NH_x$[6,9]. For MCAS, lowering of the activation energy (Ea) of adsorption for reactants ($N_2$ and $H_2$) also causes an increase in the adsorption energy ($\Delta$E) of intermediate products ($NH_x$); the result being that the catalyst's surface is quickly saturated and the overall reaction rate drops dramatically. Recent research has shown that catalytic supports including alkali and alkaline earth metal hydrides (H⁻), nitrides (N³⁻), imides (NH²⁻), amides (NH₂⁻), and nitride-hydrides (N³⁻-H⁻) can significantly promote the synthesis of $NH_3(g)$[3,5,6,8].

These N–H compounds are proposed to do this by breaking the scaling relationship between $H_2$ Ea and $NH_x$ $\Delta$E as they have a positive reaction order with respect to $H_2(g)$. The mechanism that underlies this change is widely believed to result from the transfer of H-adatoms from the TM surface to the N–H compounds[7]; in some cases N-adatoms are additionally transferred[4,6,10]. This removal of the H and N adatoms frees the catalytic surface for the reaction of $NH_x$ species into in $NH_3(g)$. Thus, good transport properties, specifically, ionic

conductivity of the support materials are crucially important for the overall heterogeneous system. Recently, several high performing hydride ion conductors (on the order of $10^{-2}$ S cm⁻¹ or better) have been reported in the literature[11–13]. Both barium hydride and the oxygen doped lanthanum hydride have been reported to show good MCAS activity[10,14], while similar oxyhydrides and oxynitride-hydrides to Ba-Li based oxyhydride of Takeiri et al. have also shown good activity[2,15,16]. This paper reports on a fast H⁻ ionic conductor: β-$Ca_2NH$, as well as establishing a nomenclature to distinguish between species of calcium nitride-hydride.

$Ca_2NH$ was first reported by Brice et al. as forming in the $Fd$-$3m$ space group with $Ca^{2+}$ ions ($32e$) forming a slightly distorted face-centred cubic close pack arrangement, and the nitride and hydride species occupying the octahedral positions ($16d$ and $16c$, respectively)[17]. A 25% intrinsic vacancy concentration due to H⁻ ions moving to the $96g$ tetrahedral position was observed. More recently, Verbraeken et al. suggested that the structure was better fitted using the $R$-$3m$ space group, which differs from the $Fd$-$3m$ in the ordering of the octahedral positions[18]. Additionally, they found a 13% intrinsic vacancy concentration at ambient temperature. The $R$-$3m$ space group is also observed for the barium and strontium analogues[19,20]. Finally,

[1]Chemistry, University of St Andrews, St Andrews, Scotland KY16 9ST, UK. [2]ISIS Neutron and Muon Source, Rutherford Appleton Laboratory, Oxford, England OX11 0QX, UK. ✉e-mail: gji4@st-andrews.ac.uk; jtsi@st-andrews.ac.uk

Verbraeken et al. noted the presence of a secondary *quasi-imide phase* that constituted approximately 10% of their system. This phase disappeared upon heating. Sichla et al. and Chemnitzer et al. observed previously that the nitride-hydrides of Sr form mixed phases containing nitride, hydride, and imide anions[20,21]. Additionally, these types of systems have been observed and studied in lithium NH compounds[22,23]. These disordered phases comes in many forms depending on the synthesis route. A paper by Weidner et al. gives a considered discussion around these *quasi-imide phases*[24]. Importantly, the disruption in site occupancy in these mixed phases posed a significant challenge for the application of these materials in reversible solid-state hydrogen storage[24]. Makepeace et al. suggest that a compositional compromise must be struck in cycling lithium imide to lithium amide during ammonia synthesis as the prior has significantly better electrical performance than the latter[23].

We are aware of four approaches to produce calcium nitride-hydride and have summarized these in Supplementary Table 2. Importantly, depending on method, the system can either contain or not contain a secondary anion species (we label these phases α- and β-respectively). Here, we show the effect the presence of the secondary species and associated loss of ordering has on structure and electrical properties of the system.

Two precursors that had been previously shown to react with $N_2$ and $H_2$ gases were chosen: a hydride[3,6,10,25] ($CaH_2$) and a nitride (α-$Ca_3N_2$)[26,27]. These compounds were heated to 600 °C under flowing 5% $H_2/D_2$ and $N_2$ in Ar respectively. The properties of the resulting phases were analysed from data collected from in situ neutron powder diffraction (NPD) collected on Polaris Diffractometer at ISIS[28] and impedance spectroscopy (using a specialized neutron rig (see Irvine et al.[29]). Details of these analyses are available in the Supplementary Methods. Next, the $CaH_2(D_2)$ system was exposed to $N_2$ gas, while the α-$Ca_3N_2$ system was exposed to $H_2/D_2$. The result in both systems was the emergence of a $Ca_2NH$ phase.

Figure 1a shows the final NPD patterns from the two experiments with quite different phases being produced. Starting from α-$Ca_3N_2$ (94 wt% with a 6% CaO impurity, see Supplementary Figure and Table 1) a fairly simple $Ca_2NH$ pattern is produced without any evidence of secondary phases. The resulting phase exhibits a pattern that is shifted towards larger d-spacings and has lost the peaks associated with the octahedral ordering in the *Fd-3m* and *R-3m* space groups (see Supplementary Figs. 2–4 and accompanying discussion). This loss of ordering allows for the indexing of $Ca_2NH$ in the rock-salt structure space group (*Fm-3m*). This space group assignment has a lattice parameter that is half the size for the analogous structure using the *Fd-*

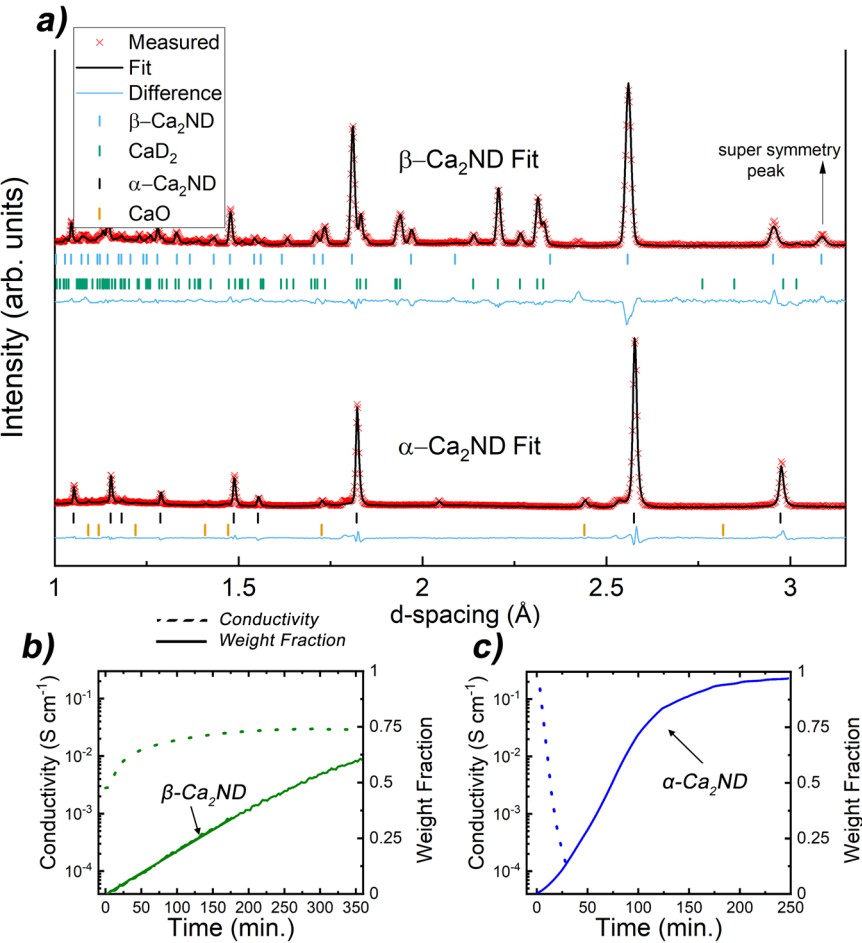

**Fig. 1 | Neutron powder diffraction (NPD) fit and conductivity results. a** NPD patterns of the α- and β- phases (from α-$Ca_3N_2$ and $CaH_2$ precursors respectively) of calcium nitride hydride(deuteride) collected on Polaris diffractometer at ISIS. The patterns show that the α-phase loses the extra peaks associates with *Fd-3m* space group. This symmetry relates to the ordering of the anionic octahedral positions. **b, c** Total conductivity of the system and weight fraction of the $Ca_2ND$ phases versus time. Solid line is weight fraction and dotted line conductivity. EIS data collected for α-phase system was done in a separate experiment that matched the conditions of the Polaris experiment. The β-phase system EIS data was collected simultaneously with the NPD data. Results show that as the α-phase emerges the measured conductivity plummets by 3 orders of magnitude while the emergence of the β-phase is associated with an increase in conductivity.

*3m* assignment. The best fit was produced by placing the $Ca^{2+}$ ions in *4a* position and $N^{3-}$ and $D^-$ (deuteride) ions in the *4b* position. Furthermore, a 20% concentration of imide ($NH^{2-}$) protons were placed in the *24e* position which correspond to proton positions in the published calcium imide structure[30]. Thus, the reaction of α-$Ca_3N_2$ with $H_{2(g)}$ at 600 °C produces a mixed/disordered phase which we label α-$Ca_2NH$. We also note that similar to other systems containing secondary anion species[18,23], the system exhibits pronounced asymmetric peaks upon cooling (see Supplementary Fig. 5). Additional experiments showed that α-$Ca_3N_2$ only begins to react to α-$Ca_2NH$ at temperatures >450 °C (see Supplementary Fig. 6 and Supplementary Table 4), and if heated to 800 °C can be fully converted to CaNH (see Supplementary Fig. 7 and Supplementary Table 5).

There are a large number of extra peaks present in the sample produced from $CaD_2$ (pure at the start of the experiment) as a result of a phase equilibrium between the produced $Ca_2ND$ phase and the N-doped $CaD_2$ precursor (see Supplementary Figs. 8, 9, and Supplementary Table 6). Whilst the residual content is decreasing with time, its content is not tending to zero, hence an equilibrium mixture is being approached.

The $Ca_2ND$ phase does not show the disordering observed in an α-phase and was indexed and fitted using the *Fd-3m* space group and model proposed by Brice et al.[17], nor does it show any evidence of a secondary anion species and thus the phase is labelled β-. The relative lattice parameters for two phases are 10.3001(12) Å vs 10.23062(24) Å for the α- and β-phases, respectively. Previous research has shown that solid solutions of NH compounds have larger lattice parameters than the pure compounds in agreement with the findings here[24,31,32].

The β-phase agrees well with the result published by Brice et al.[17]. Full refinement yielded an occupancy of the D1 octahedral site of 0.822(11) as opposed to the value of 0.75 reported by Brice et al., with the remainder being located in a tetrahedral site (18.9%, as opposed to 25% previously published) giving a stoichiometric compound ($Ca_2NH_{0.997(34)}$). One major difference between the fit in this paper and that of Brice et al. is the position of the D2 site. It was found that refinement of the *96g* position proposed by Brice et al. was unstable and required that the atomic displacement parameter (ADP), occupancy, and coordinates be fixed. Using the *48f* position instead allowed for the full refinement of the position, although the low occupancy and high multiplicity create larger than normal errors. A comparison between the fit results of the two phases is shown in Table 1. Further experiments revealed that the synthesis of $Ca_2NH$ from $CaH_2$ required temperatures above 400 °C. Additionally, heating β-$Ca_2NH$ to 800 °C resulted in the appearance of shoulder peaks (in a post experiment powder x-ray diffraction pattern, see Supplementary Fig. 7) associated with secondary anionic species.

Next, we explore the effect the order versus disorder has on the electrical properties of the systems.

Electrochemical Impedance spectroscopy (EIS) data collected during the reaction of $CaD_2$ with $N_2$ showed that as the experiment proceeded, measured conductivity increased by an order of magnitude from $3 \times 10^{-3}$ S cm$^{-1}$ to $3 \times 10^{-2}$ S cm$^{-1}$ (see Fig. 1b). The rise in conductivity observed correlated with the increasing content of the β-$Ca_2ND$ phase. A separate experiment found that doping of $CaH_2$ with $N_2$ resulted in an increase in measured conductivity from $5.63 \times 10^{-3}$ S cm$^{-1}$ to $5.8 \times 10^{-2}$ S cm$^{-1}$ while the post-experiment x-ray diffraction pattern showed a 9:1 ratio of β-$Ca_2NH$ to $CaH_2$ (see Supplementary Fig. 10). Previous work also showed that the precursor $CaH_2$ had an ionic conductivity on the order of $10^{-3}$ S cm$^{-1}$ at 600 °C[33]. The difference in conductivities would influence ammonia production where hydrogen flux from the hydride becomes limiting, noting that in the systems shown by Hattori et al.: $CaH_2$ and $Ca_2NH$ mixed with Ru at 340 °C produced ammonia at similar rates where the better conducting $Ca_2NH$ system had significantly less Ru catalyst by weight[3].

For the α-phase, a second experiment conducted at the University of St Andrews that replicated the experimental conditions of the Polaris experiment (with $H_2$ instead of $D_2$) found that the upon reaction of α-$Ca_3N_2$ with hydrogen, the measured conductivity plummeted by 4 orders of magnitude (from $10^{-1}$ S cm$^{-1}$ to $10^{-4}$ S cm$^{-1}$, see Fig. 1c). Note that α-$Ca_3N_2$ showed good conductivity properties in accord with electronic semiconductor behaviour (see Supplementary Fig. 11). Thus, the EIS data shows that the α-phase has poor electrical properties, while the β- is a potential fast ion conductor, similar to previously published results for $Ba_2NH$[19,34].

To further explore the structural difference between the two phases, NMR and Raman spectra were collected at room temperature. Figure 2a, b shows the NMR and Raman, respectively. The NMR $^1H$ peak at 5.2 ppm is relatively broad for the α-phase compared to the β-. Broadening of this type can result from dynamic motion or static disorder[35]. In this case, it is likely the result of static disorder as the spectra were collected at room temperature. The relative disorder of the α- mirrors the results from the NPD study that showed that the α-phase octahedral sites lacked the ordering found in the β-phase causing a change in symmetry and formation in the rock salt structure. Furthermore, the α-phase NMR shows the existence of secondary hydrogen species associated with amide ($NH_2^-$) at 1.0 ppm and imide at ($NH^{2-}$) at −2.7 ppm. The amide appears more prevalent in the spectrum due to the fact that its peak is quite sharp compared to the imide which is not only positionally distributed around the bonded N atom, but also randomly distributed throughout the host lattice. The final peak at −1.3 ppm is thought to be associated with OH$^-$ groups formed during loading and unloading the sample into the NMR instrument.

Raman data also show broadening in the α-phase spectrum for the peaks associated the nitride-hydride species (292 cm$^{-1}$)[7]. This peak is also shifted to lower wavenumbers as compared to the β- (328 cm$^{-1}$) suggesting that the configuration of the α-phase is lower energy. This result matches well with published results on disordered lattices[22, 36]. The α-phase also shows the characteristic broad peaks between 100 and 1000 cm$^{-1}$ of the imide species, as well as the sharp peak at 3125 cm$^{-1}$[7,22].

Together, the NMR and Raman data confirm the NPD data structure refinement results that the α- exhibits disorder and contains a concentration of imide species while the β- is relatively ordered at room temperature with a single NMR signal implying thermally induced defects.

The β-phase conductivity was explored further on Osiris spectrometer at ISIS[37]. A pellet of $CaH_2$ was loaded into the specialized in situ rig. The sample was exposed to short 5 min doses of $N_2$ in Ar and $H_2$ (5/90/5 cm$^3$ min$^{-1}$ respectively) at 600 °C. NPD patterns were collected before and after each dose, while EIS and quasi-elastic neutron scattering (QENS) data were collected over the course of several hours. Further details and example data and fits are available in Supplementary Figs. 12–14. Data sets are labelled *N* followed by a numeral representing the dose number, while the number after the hyphen is the dataset for that dose.

The first measurable broadening of the quasi-elastic data came after the 4th nitrogen dose and the second QENS dataset (labelled N4-2 in Fig. 3). This dose also saw the first significant rise in the phase fraction of β-$Ca_2NH$ as measured by NPD as well as a jump in the measured ionic conductivity as measured by EIS ($\sigma_{EIS}$) (Fig. 3). Thus, the β-$Ca_2NH$ phase is again correlated with the rise in $\sigma_{EIS}$. Additionally, the associated QENS broadening strongly suggest that the increase in $\sigma_{EIS}$ is associated with ionic diffusion of hydrogen, in this case likely H$^-$ ions. The QENS data were modelled by the Chudley-Elliott jump diffusion model (CEM)[38] which produces two important physical characteristics of the diffusion process: the jump length ($l$), and the residence time ($\tau$). In order to understand the mechanism of diffusion it is important to contrast the extracted $l$ with site to site distances. Figure 4 shows a plot of different site to site distances from the β-$Ca_2NH$ phase refinement of the Osiris diffraction patterns as well as the

**Table 1 | Comparison between α- and β-phase fits from neutron powder diffraction data**

| NPD Data @ 600 °C | α-Ca₂ND (*Fm-3m*) [Ca₂N₀.₈D₀.₈(ND)₀.₄] | | | | | | |
|---|---|---|---|---|---|---|---|
| a (Å) | 5.15005 (6) | Ca (0,0,0) | | D1 (½,½,½) | | D2 (*x,y,z*) -imide proton | |
| Volume (Å³) | 136.5945 (28) | Fraction | 1ᶠ | Fraction | 0.4ᶠ | *x* | 0 |
| | | Uiso x100 (Å²) | 3.317 (30) | Uiso x100 (Å²) | 2.258 (20) | *y* | 0 |
| wRp | 0.0288 | | | N1 (½,½,½) | | *z* | 0.35ᶠ |
| Rp | 0.0412 | | | Fraction | 0.6ᶠ | Fraction | 0.0333ᶠ |
| χ² | 7.262 | | | Uiso x100 (Å²) | 2.921 (20) | Uiso x100 (Å²) | 3.00ᶠ |
| | β-Ca₂ND (*Fd-3m*) [Ca₂ND₀.₉₉₇] | | | | | | |
| a (Å) | 10.23062 (24) | **Ca (*x,x,x*)** | | **D1 (0,0,0)** | | **D2 (*x,y,z*) -hydride** | |
| Volume (Å³) | 1070.79 (4) | *x* | 0.26171 (11) | Fraction | 0.808 (10) | *x* | 0.125ᶠ |
| | | Fraction | 1ᶠ | Uiso x100 (Å²) | 5.73 (16) | *y* | 0.125f |
| wRp | 0.0177 | Uiso x100 (Å²) | 1.73 (4) | N1 (½,½,½) | | *z* | −0.0904 (30) |
| Rp | 0.0308 | | | Fraction | 1ᶠ | Fraction | 0.063 (11) |
| χ² | 4.889 | | | Uiso x100 (Å²) | 2.030 (29) | Uiso x100 (Å²) | 5.39 (53) |

The α-phase is associated with loss of symmetry in the anionic sublattice. Furthermore, the two phases have distinctly different secondary *D* sites; the α- being characteristic of a imide proton, while the β- is a tetrahedral coordinated hydride site. The imide concentration of the α- phase is 20%, while the tetrahedral site of the β-phase has a 18.9% concentration. The actual stoichiometry of the phases are given in brackets. Parameters fixed for the refinement are denoted with a superscript f. Standard errors are reported in parentheses. Additional fitting statistics available in Supplementary Table 3.

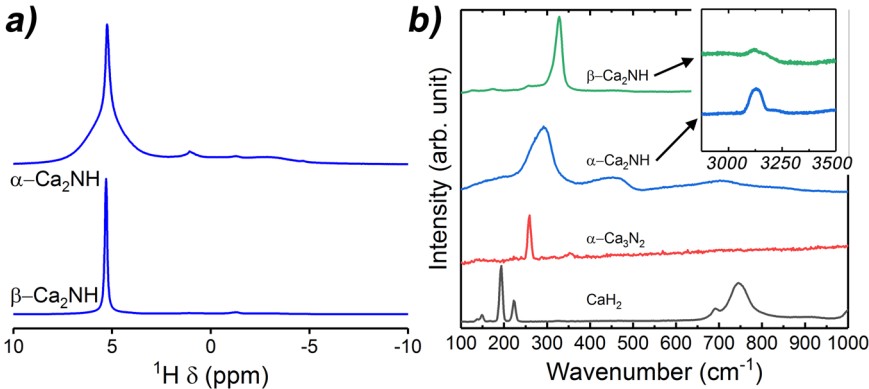

**Fig. 2 | NMR and Raman spectra for the Ca₂NH phases. a** NMR results collected on the α- and β-phases of Ca₂NH. Spectra show the ordering difference between the two phases that likely results from the presence of a secondary anion species. This disordering is manifested as a broadened H⁻ signal (5.2 ppm) at room temperature as well as the presence of other peaks associated with amide (1.0 ppm) and imide species (−2.7 ppm). **b** Raman Results collected on precursors and α- and β- phases of Ca₂NH. The spectra also show that the main band associated with the nitride-hydride at ~300 cm⁻¹ is broadened and shifted to lower energies for the α-phase. Furthermore, the presence of a peak at 3125 cm⁻¹ confirms the presence of imide protons in the α-phase. There is also a small peak for the β- at this wavenumber but it is relatively weak implying that the relative concentration is low.

*l* extracted from the CEM. The results show that the *l* values are close to the octahedral site to site distances. Therefore, conductivity values were calculated for a H1-H1 pathway using the Einstein Diffusion equation in conjunction with the Nernst-Einstein equation:

$$D_{QENS} = \frac{l^2}{n\tau} \qquad (1)$$

$$\sigma_{QENS} = D_{QENS}\frac{n_i (z_i e)^2}{k_B T} \qquad (2)$$

Where, *n* is 2x dimensionality of the diffusion process (e.g. one dimensional, two dimensional, or three dimensional), $n_i$ is the number density of species *i* (atoms cm⁻³), and $z_i e$ is the charge of species *i*. The number densities were calculated from the NPD structure refinements

using the following equation:

$$n_i = \frac{m_i O_i}{V_{cell}} \qquad (3)$$

Where $m_i$ and $O_i$ are the multiplicity and occupancy of a species in site *i*, and $V_{cell}$ is the volume of the unit cell. The resulting $\sigma_{QENS}$ values are significantly higher (~0.5 S cm⁻¹ for 3D diffusion, and ~0.7 S cm⁻¹ for 2D) than the measured $\sigma_{EIS}$ (~0.08 S cm⁻¹). However, it was noticed that if the $n_{H2}$ (the number density of the tetrahedral site, see Supplementary Fig. 15) was used instead, then the conductivity values were much closer (~0.098 S cm⁻¹). Since, the $n_{H2}$ site is associated with intrinsic anti-Frenkel defects, $n_{H2}$ should reflect the vacancy concentration in the octahedral position ($n_{v,H1}$) for this β-Ca₂NH structure which refines as overall stoichiometric. Thus, a modified Nernst-Einstein equation that takes into account the fraction of charge carrier sites that have

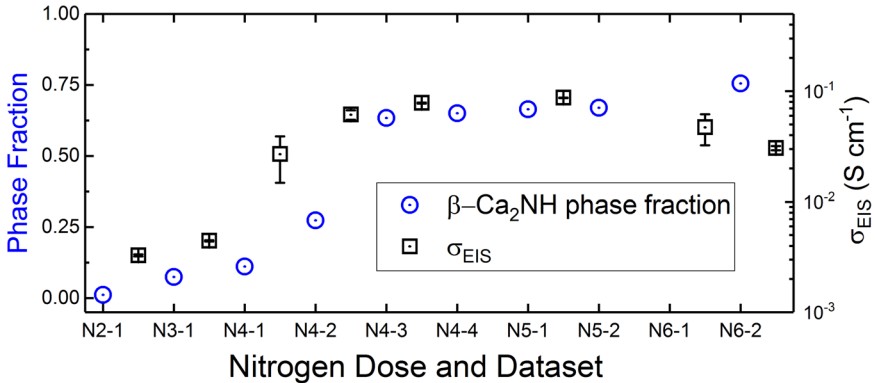

**Fig. 3 | Ionic conductivity ($\sigma_{EIS}$) and phase fraction versus nitrogen doping.** The data show that as the phase fraction of the nitride-hydride grows, so does the $\sigma_{EIS}$. The $\sigma_{EIS}$ are offset to show that the values represent the average of the measured conductivities between diffraction experiments. The error bars represent one standard deviation. Notice that after dose 4 and the first diffraction dataset that a large error is present for $\sigma_{EIS}$ This error disappears when the phase fraction stabilizes sometime after the second diffraction pattern was taken (N4-2). The large change is EIS signal is shown in Supplementary Fig. 14. The next important

information in this figure is the drop in conductivity after the 6th dose. Upon the 6th dose of nitrogen the resistance associated with the sample−electrode interface grew dramatically probably a result of delamination. This resulted in the Rs value associated with the bulk conductivity being difficult to measure accurately (see Supplementary Fig. 14); allowing for only the first few datasets to be analysed. EIS results obtained in a separate experiment show that the $\sigma_{EIS}$ is stable for β-Ca$_2$NH over long periods (see Supplementary Fig. 16).

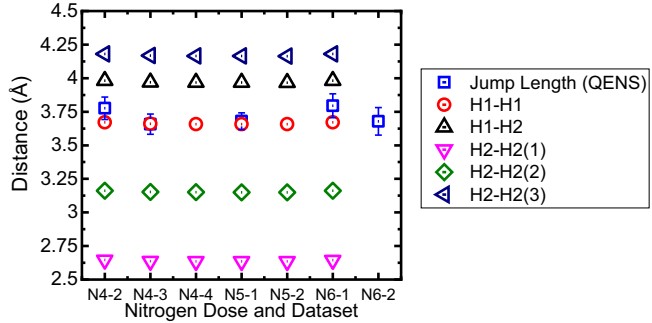

**Fig. 4 | Jump length versus lattice distances.** A plot of jump length ($l$) extracted from the quasi-elastic neutron scattering data via the Chudley–Elliott model[38] versus H$^-$ site to site distances for β-Ca$_2$NH at 600 °C. Results show that the jump length is close to the H1 to H1 site to site distance implying that the diffusive motion takes place via octahedral site to octahedral site jumps. Error bars represent one standard deviation.

neighbouring vacancies ($\chi_{i,vac}$) was employed:

$$\sigma_{QENS} = D_{QENS} \frac{n_i (z_i e)}{k_B T} \chi_{i,vac} \qquad (4)$$

There are two equivalent ways to think about the effect of this term. First, one can consider an *effective* charge carrier concentration:

$$n_i \chi_{i,vac} = n_{i,eff} \qquad (5)$$

In this case, $n_{i,eff}$ defines the number of atoms in site $i$ that have a neighboring vacancy and thus are able to diffuse. The second way is to define $D_H$ (diffusion coefficient for H$^-$) from $D_{QENS}$ (the diffusion coefficient as measured by QENS). Since QENS broadening only occurs due to dynamic motion, the QENS technique is only sensitive to the individual atoms that are mobile (either locally or over long range). In this case, the mobile atoms are the atoms with neighbouring vacancies which means that QENS is actually measuring the diffusion coefficient of the vacancies ($D_{vac}$). This term can be related to the diffusion coefficient of H$^-$ ($D_{H^-}$) via the

following relations:

$$D_{i,vac} n_{i,vac} = D_i n_i \qquad (6)$$

$$\frac{n_{i,vac}}{n_i} = \chi_{i,vac} \qquad (7)$$

$$D_{i,vac} \chi_{i,vac} = D_i \qquad (8)$$

Hence, Eq. 4 is modified to be either expressed as function of $D_{H^-}$ or as a function of $n_{i,vac}$. The NPD structure refinements returned a vacancy concentration of 18.9% ($\chi_{vac} = 0.189$) for β-Ca$_2$NH. When this equation is applied to the H1 sites, the agreement between the $\sigma_{QENS}$ and $\sigma_{EIS}$ becomes remarkable (see Fig. 5 insert). The error associated with the refinement of the occupancy was just 1.3%. Thus the combined QENS and EIS data point to a vacancy mediated conductivity pathway for β-Ca$_2$NH. Furthermore, the results show that a 3D model results in a significantly better fit than a 2D model indicating that the arrangement of octahedral sites agrees with the *Fd-3m* space group, rather than the 2D hydride array predicted by *R-3m*. This result also explains why the α-phase has poor ionic conductivity: there are no vacancies in the octahedral sites likely as a result of the entropic contribution of the imide species (i.e. the observed broadness of the NMR peak at room temperature for the α-). A similar result is seen in the difference in conductivity between lithium imide (Li$_2$NH, *Fd−3m*) and lithium amide (LiNH$_2$, I-4) where the former showed good conductivity properties due to the creation of 2 Frenkel pair defects[39,40], similar to what we see here.

Comparing the results of this combined in situ QENS and EIS experiment with previously published research on BaH$_2$ shows that the $D_{QENS}$ measured for the β-Ca$_2$NH system is higher than that measured for BaH$_2$[29]; while the overall conductivity at 600 °C for the two systems in significantly higher for BaH$_2$−the difference arising from the significantly higher effective charge carrier concentrations in the BaH$_2$ due to the presence of a concerted migration mechanism. Table 2 summarizes the difference between the two systems. Thus, while both these systems are fast hydride ion conductors, the mechanisms by which they achieve this are vastly different.

We have presented the H$^-$ ion conductor β-Ca$_2$NH and also shown that the closely related α-phase (containing disorder due to the presence of secondary anionic species) has poor ionic conductivity

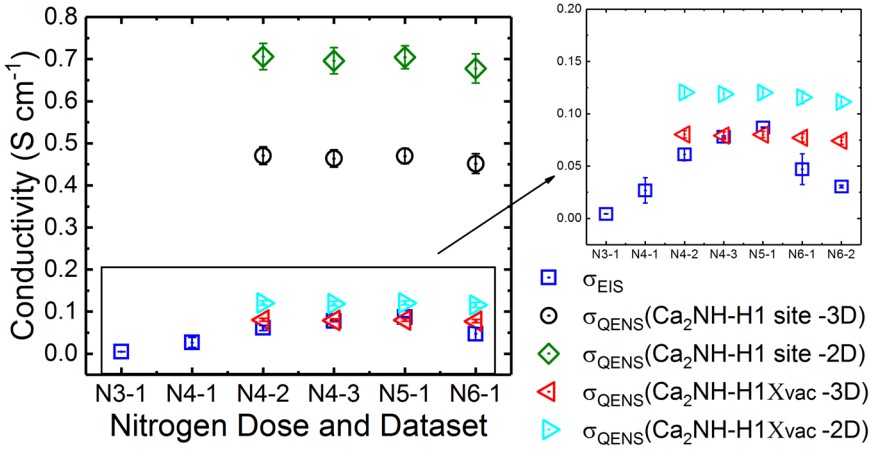

**Fig. 5 | Conductivity comparison from different techniques.** The results show that a 2D and 3D pathway for the H1 site results in a predicted ionic conductivity for the quasi-elastic neutron scattering (QENS) data that is significantly higher than that measured by impedance spectroscopy (EIS) (0.08 S cm⁻¹ vs 0.7 S cm⁻¹ for 2D and 0.5 S cm⁻¹ for 3D). However, if the probability of the a 1H site having a neighboring vacancy is taken into account (see Eqs. 4–6) then the 3D pathway QENS calculated value agree remarkably well with the EIS measured conductivity (for N5−1, 0.080 S cm⁻¹ for QENS versus 0.084 S cm⁻¹ for EIS). This result implies a vacancy mediated diffusion mechanism. It is believed that difference between the techniques for the 6th nitrogen dose arises due to issues with the electrode–sample interface. This explored further in Supplementary Fig. 14. Error bars represent one standard deviation.

**Table 2 | A comparison between two hydride ion conductors**

| System @ 600 °C | $n_{H^-}$ (cm⁻³) | $\sigma$ (S cm⁻¹) | D (cm² s⁻¹) |
|---|---|---|---|
| $BaH_2$ | $1.50\,(2) \times 10^{22}$ | 0.32 (3) | $1.17\,(3) \times 10^{-5}$ |
| $\beta$-$Ca_2NH$ | $2.31\,(5) \times 10^{21}$ | 0.0782 (3) | $1.63\,(6) \times 10^{-5}$ |
| ratio | 6.5 | 4.0 | 0.72 |

Although, calcium nitride-hydride has a higher diffusion coefficient than barium hydride[29], its overall conductivity is significantly lower due to the almost 10x lower effective charge carrier concentration. The dramatic difference in this concentration is due to the different driving mechanisms for diffusion. Standard Error are shown in parentheses and correspond to the smallest magnitude numeral.

potentially as a result of the lack of intrinsic vacancies caused by the presence of the imide species in the phase; this result has important implications for MCAS in that the ionic conductivity of the support material may play a fundamental role in promoting the TM catalyst via the removal of adatoms from the surface. This is evinced by the Mars-van Krevelen mechanism observed by Kitano et al. and by the chemical looping of $BaH_2$ systems by Gao et al.[4,7], both implying significant contributions to the overall process from the bulk of the N–H support materials.

This paper has provided direct evidence for a vacancy-mediated conductivity pathway by using combined analysis of simultaneously collected QENS and EIS data. Finally, this paper has shown that secondary characteristic techniques such as NMR and Raman are important to verifying and understanding the crystal structure of N–H materials.

## Methods
All materials were handled under Ar atmosphere, as the samples are air-sensitive. Under these conditions, the phases are stable.

### Neutron powder diffraction
$CaD_2$ was synthesized from pure calcium shot in a sealed reactor at 800 °C for 6 h under flowing 5% $D_2$ in Ar at 100 cm⁻³ min⁻¹. 25 mm pellets were pressed of $CaD_2$ and $\alpha$-$Ca_3N_2$ (purchased from Alfa Aesar >99%) and sintered at 800 °C for 2 h under flowing 5% $D_2$ / $N_2$ respectively in Ar at 100 cm⁻³ min⁻¹. The sintered pellets were then painted with palladium ink (C2031105P2) from Gwent Group and the electrodes were dried inside an oven in a glovebox at 100 °C for 10 min.

A microscopic camera was used to photograph the electrodes along with a calibration slide. These pellets were loaded into a specialized in situ cell (the St Andrews in situ cell) at ISIS inside a glovebox and then placed inside the RAL4 furnace loaded on Polaris[28]. All diffraction data structures were refined using GSAS[41].

### Quasi-elastic neutron scattering experiment
$CaH_2$ was purchased from Fluka. A 25 mm pellet of $CaH_2$ was prepared in identical manner as the NPD experiment (with the substitution of $H_2$ for $D_2$) except the pellet was extremely thin to allow for the 90% transmission rule (e.g. a rule of thumb to reduce signal to noise ratio). The St Andrews in situ rig was used again but this time in conjunction with the RAL3 furnace and the Osiris spectrometer at ISIS[37]. Simultaneous EIS, diffraction, and QENS data were collected. The sample was oriented at a 45° angle to the incoming neutron beam to maximize the exposed sample area. The sample was heated to 600 °C under flowing Ar and $H_2$ (95 and 5 cm⁻³ min⁻¹ respectively). A Solartron 1280Z was used to collect EIS data with a 40 mV perturbation voltage between 20 kHz and 1 Hz.

### NMR and Raman experiments
$\alpha$-$Ca_2NH$ and $\beta$- were synthesized at St Andrews at 600 °C with exposure to 5/5/90 cm⁻³ min⁻¹ $N_2$/ $H_2$/ Ar for the nitride and hydride precursors. These materials were loaded into air-tight sample holders inside a glovebox and then transported to the NMR and Raman facilities. The NMR spectra were collected on a 9.4 T spectrometer with a magic angle spin of 14 kHz. The Raman spectra were collected using Renishaw inVia Raman Microscope with a 532 nm laser at 5% power collected over 10 s from 100 cm⁻¹ to 3500 cm⁻¹.

## Data availability

The research data underpinning this publication can be accessed at https://doi.org/10.17630/6760d05d-114b-47ee-99bf-c9481cf25a55. For neutron data see refs. 42–44.

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

## Acknowledgements

Neutron scattering beamtime (RB1720395[42], RB1410181[43], RB1720393[44]) at the ISIS neutron and muon source was provided by the UK Science and Technology Facilities Council (STFC). Prof John TS Irvine and Prof Martin Owen Jones: STFC 5005—Development of Combined In situ Neutron Diffraction and Electrochemical Studies. We thank Dr Daniel Dawson for assistance with Solid State NMR data collection at St Andrews.

## Author contributions

G.J.I.: manuscript, data collection and analysis. R.I.S.: Neutron diffraction data experiments setup and collection/reduction. M.O.J.: main author's academic advisor, and conceptualisation. J.T.S.I.: main author's academic advisor, editing and conceptualisation.

## Competing interests

The authors declare no competing interests.
