## [Peer Review File · Nature Communications]

Order-Disorder and Ionic Conductivity in Calcium Nitride-Hydride Ammonia Synthesis CatalystsREVIEWER COMMENTS

Reviewer #1 (Remarks to the Author):

This manuscript reports the hydride-ion conductivity of calcium nitride-hydride and related phases. The authors identify two different polymorphs of calcium nitride-hydride based materials, and contrast the conductivity of the two on the basis of the influence of disorder (structural and composition-based) on the conductivity. Their detailed in situ diffraction, impedance spectroscopy and quasi-elastic neutron scattering provides a comprehensive description of the conductivity of the phases.

As the authors note, Ca-N-H phases have been used in conjunction with transition metals to form some of the most interesting catalyst formulations for ammonia synthesis under mild conditions. Indeed, the reversible incorporation of H species in these phases has been identified as one of the features behind their proposed positive impacts on the catalytic activity. As such, a thorough investigation of these phenomena will be of significant interest to the catalyst community.

The work has been carefully completed and the results support the discussion in the manuscript. I have provided a number of specific points of feedback to the authors below. One general point is that, aside from in the introduction, there is very little connection drawn between the results obtained and the catalytic data which has been previously collected on Ca-N-H supports. Given the general audience of this journal, I would suggest that further effort should be made to guide readers to the implications of the conductivity results obtained on the veracity of the catalytic mechanisms proposed for these materials. For example, catalytic measurements have been made on CaH₂, Ca₂NH, CaNH and Ca(NH₂)₂, and it would be helpful to comment on the contrast in catalytic activity in light of the data collected. This would significantly enhance the impact of the manuscript and in my opinion is required if this is to be published.

Some specific points of feedback:

- I would propose that the authors' description of the Fm-3m phase as alpha-Ca₂NH be given some further context. Given the 20% imide content refined in this structure, could they provide further justification for describing it as a polymorph of calcium nitride-hydride rather than a solid solution of Ca₂NH and CaNH, given the structural similarities they highlight. This distinction may help strengthen their arguments for the importance of a well-defined composition to achieving high conductivity later in the work.

-The discussion of the influence of the synthesis conditions on the formation of an imide-contaminated sample of Ca₂NH would be helpful, particularly whether reactions of H₂/N₂ at different stages might be causing the formation of imide groups.

-The description of scaling relations in the introduction refers to absorption energies. In the context it is presented, it might be clearer to describe these as adsorption energies.

-When the authors report the conductivity values for the phases, could they put these in context with other hydride conductors. Are they at a level which might explain their interesting catalytic properties compared with other materials?

-The description of the difference in conductivity of lithium amide and imide is oversimplified, since it is lithium amide which can be described as having lithium site vacancies compared with lithium imide. Vacancies are less relevant in that system since Frenkel defect formation is thought to govern the conductivity.

-The detail of the Raman experiments is missing from the Experimental section.

-Figure 1 has some text cut off at the side. Parts b and c also do not make clear the different reactions - could the authors add the weight fractions of the starting materials as well for clarity?

-The layout of Table 1 is visually confusing. Would it be possible to have the lattice parameters

and goodness of fit details in separate rows to the site details?

-There are a number of formatting inconsistencies throughout the manuscript and SI which need to be addressed (e.g. capitalisation, chemical formulae, introduction of QENS acronym)

Reviewer #2 (Remarks to the Author):

The manuscript "Order-Disorder and Ionic Conductivity in Calcium Nitride-Hydride Ammonia Synthesis Catalysts" presents results on two phases of Ca_2NH , one of which shows reasonably high hydride ionic conductivity. In this respect, the work builds on recent developments in the solid-state hydride conductor literature, particularly with regard to mixed-anionic conductors (oxyhydrides, nitride hydrides) that have attracted recent interest.

The data collection and analysis are thorough and appear to be well documented for reproducibility, though the discussion and presentation are at times confusing. It is not always clear whether the authors have justifications for some of the assumptions they make. In several cases, computational analysis would be an appropriate means to support the authors' claims. Among the comments that caught my attention:

On pg 6: "Since, the nH_2 site is associated with intrinsic anti-Frenkel defects, nH_2 is equal to the vacancy concentration in the octahedral position ($\text{n}_{\text{v,H1}}$)." This seems to assume two things: one, that there is an exact stoichiometric concentration of hydrogen in the system, and two, that hydrogen can reside solely on the octahedral or tetrahedral sites (not, for instance, on nominal N sites). Can these assumptions be justified? If there are lots of H atoms on tetrahedral sites, it seems suspicious that there apparently aren't many H1-H2 jumps contributing to the measured conductivity.

Of α - Ca_2NH , on pg 6: "there are no vacancies in the octahedral sites likely as a result of the entropic contribution of the imide species." I assume that, in a perfect structure, all H in α - Ca_2NH would reside on octahedral sites. If that is the case, full octahedral occupation ("no vacancies", as the authors say), plus protonic interstitials (the imides), would add up to significant excess hydrogen. Does α - Ca_2NH contain a significant H excess?

On pg 7, comparing Ca_2NH to BaH_2 : "the difference arising from the significantly higher charge carrier concentrations in the BaH_2 due to the presence of a concerted migration mechanism." Previous work, including reports from some of the same authors, has shown clearly that BaH_2 has a vacancy-mediated conduction pathway, which seems analogous to that reported here for Ca_2NH . Thus, invoking a different "migration mechanism" to explain the difference in conductivities does not make sense to me.

The caption of Fig. S4 states: "It is expected from thermal vibrations that peaks would broaden with increasing temperature. In this case, we see the opposite trend." Can the authors explain this discrepancy?

Additional information regarding the stability of these structures is also requested. Specifically, in light of the observation of SrO in several diffraction plots, how stable are they in ambient conditions? This information is important for further experimental development. And which of the two phases in question (alpha or beta) is thermodynamically preferred?

The presentation of the structures is also confusing. In place of, or in addition to, the text description of the phases, I recommend modifying Fig S1 to show explicitly the crystal structures under consideration and label them as α - Ca_2NH and β - Ca_2NH . Including the R-3m phase doesn't seem necessary, since it doesn't participate in the authors' discussion, but if it is discussed, it should be assigned its own Greek letter for clarity. Also, please indicate the H1 and H2 sites, which are invoked in discussions of H transport.

Addressing these concerns is necessary for me to recommend this work as a contribution to Nature Communications.

Reviewer #3 (Remarks to the Author):

This paper reports on Order-Disorder and Ionic Conductivity in Calcium Nitride-Hydride Ammonia Synthesis Catalysts. The paper is well written and easy to follow, and the subject is relevant. The approach used is valid and the conclusions being drawn from results are correct.

The manuscript can be accepted after the following minor revisions:

1) The notation of Space Group is not correct: please change Fd-3m and R-3m using italic for Lattice and mirror plane

2) Wyckoff letter (i.e. 96g, 48f) must be italic (96g, 48f).

3) The authors should always specify:

- Radiation type and wavelength.
- Data collection conditions: 2θ range, step size, time per step;
- The software used for the Rietveld refinement.
- The structural models used for the full-profile fit refinement (giving the sources: literature reference, database or phase code);
- If a correction for preferred orientation effects (e.g., March-Dollase function, or spherical harmonics) has been applied.
- No. of variables,
- No. of observations
- No. of reflections
- R_p (%), R_{wp} (%) and RF^2 (%)
- residual electron density (min/max $e\text{\AA}^{-3}$)

4) Page 4, please report the Standard errors of unit cell parameters in parentheses ("The relative lattice parameters for two polymorphs are 10.30 Å vs 10.23 Å for the α - and β -phases respectively....).

5) Table 1: please comment the very high standard errors of Uiso for D2 (3.17(372) and 5.39(176)). These value are not acceptable. The refinement of these parameters must be revised.

Reviewer #4 (Remarks to the Author):

This paper reports order-disorder and ionic conductivity in Ca-N-H ammonia catalysis.

Distinct hydride conductivity difference between the ordered (beta-phase) and disordered phase (alpha phase) in Ca₂NH was experimentally clarified and the microscopic mechanism on high conductivity was well explained on the data evaluated from QENS measurement.

However, the claim on the catalyst for NH₃ is too speculative at the present stage:

First, the catalytic activity of transition-metal unloaded Ca₂NH is not so high. Thus, Ca₂NH is not the main catalyst but may be regarded as co-catalyst.

Second, the activation of energy of NH₃ synthesis is much higher than that of hydride ion conduction as reported in Adv.Energ. Mat. 11, 2003723(2021). This fact means the rate determining step of the NH₃ synthesis is not ionic diffusion. It is likely the formation of low work function electron at the site of hydride vacancy plays a crucial for N₂ activation through electron donation to loaded transition metal. So, hydride vacancy works well for this purpose. This model is supported by a striking difference in the catalytic activity between Ca₂NH and CaNH (Chem.Sci. 7, 4036(2016)).

The authors should confine the main claim which can be supported by the solid experimental results.

Minor comments:

1. No QENS data were shown and experimental description is too brief.
2. There are several types such as Table S1 (3rd column, Ca₂NH + 1/2 N₂ → Ca₂NH, N₂ → H₂) and incomplete sentences in the text.

RESPONSE TO REVIEWERS' COMMENTS

We would like to thank each of the the reviewers for taking the time to carefully consider our manuscript and for providing us with helpful feedback.

We have endeavoured to answer all the questions raised and address any concerns. We hope that these changes meet approval.

Many thanks

Reviewer #1 (Remarks to the Author):

This manuscript reports the hydride-ion conductivity of calcium nitride-hydride and related phases. The authors identify two different polymorphs of calcium nitride-hydride based materials, and contrast the conductivity of the two on the basis of the influence of disorder (structural and composition-based) on the conductivity. Their detailed in situ diffraction, impedance spectroscopy and quasi-elastic neutron scattering provides a comprehensive description of the conductivity of the phases.

As the authors note, Ca-N-H phases have been used in conjunction with transition metals to form some of the most interesting catalyst formulations for ammonia synthesis under mild conditions. Indeed, the reversible incorporation of H species in these phases has been identified as one of the features behind their proposed positive impacts on the catalytic activity. As such, a thorough investigation of these phenomena will be of significant interest to the catalyst community.

The work has been carefully completed and the results support the discussion in the manuscript. I have provided a number of specific points of feedback to the authors below. One general point is that, aside from in the introduction, there is very little connection drawn between the results obtained and the catalytic data which has been previously collected on Ca-N-H supports. Given the general audience of this journal, I would suggest that further effort should be made to guide readers to the implications of the conductivity results obtained on the veracity of the catalytic mechanisms proposed for these materials. For example, catalytic measurements have been made on CaH_2 , Ca_2NH , CaNH and $\text{Ca}(\text{NH}_2)_2$, and it would be helpful to comment on the contrast in catalytic activity in light of the data collected. This would significantly enhance the impact of the manuscript and in my opinion is required if this is to be published.

We have noted at the end of the NPD discussion (pg 4) The difference in conductivities would influence the rate of ammonia production where hydrogen flux from the hydride becomes limiting, noting that CaH_2 and Ca_2NH have been compared for ammonia synthesis rates by Hattori et al. and found to produce ammonia at similar rates at 340 °C even though the weight % of Ru is significantly lower for the better conducting nitride-hydride system.

Some specific points of feedback:

- I would propose that the authors' description of the Fm-3m phase as alpha- Ca_2NH be given some further context. Given the 20% imide content refined in this structure, could they

provide further justification for describing it as a polymorph of calcium nitride-hydride rather than a solid solution of Ca_2NH and CaNH , given the structural similarities they highlight. This distinction may help strengthen their arguments for the importance of a well-defined composition to achieving high conductivity later in the work.

We agree that the use of polymorph does not benefit the discussion and have changed this to phase (e.g. a-polymorph to a-phase) and added language to distinguish between a- and b- (with or without a secondary anionic species). Whilst it is useful to consider this as a solid solution, there is an important degree of local ordering which might not be conveyed by the term solid solution.

-The discussion of the influence of the synthesis conditions on the formation of an imide-contaminated sample of Ca_2NH would be helpful, particularly whether reactions of H_2/N_2 at different stages might be causing the formation of imide groups.

We added details describing a set of experiments that explore the effects of temperature on the formation of the imide secondary species from both precursors. The resulting phases do have an important degree of metastability and so under kinetic control. A full exploration of the effects of N/H_2 partial pressures and temperature will make an interesting topic for future experiments.

-The description of scaling relations in the introduction refers to absorption energies. In the context it is presented, it might be clearer to describe these as adsorption energies.

Changed.

-When the authors report the conductivity values for the phases, could they put these in context with other hydride conductors. Are they at a level which might explain their interesting catalytic properties compared with other materials?

We have added the following to the introduction:

“Recently, several high performing hydride ion conductors (on the order of 10^{-2} S cm^{-1} or better) have been reported in the literature^{11–13}. Both barium hydride and the oxygen doped lanthanum hydride have been reported to show good MCAS activity^{10,14}, while similar oxyhydrides and oxynitride-hydrides to Ba-Li based oxyhydride of Takeiri et al. have also shown good activity^{2,15,16}.”

-The description of the difference in conductivity of lithium amide and imide is oversimplified, since it is lithium amide which can be described as having lithium site vacancies compared with lithium imide. Vacancies are less relevant in that system since Frenkel defect formation is thought to govern the conductivity.

We have taken this on board and have clarified the language with reference to these systems to match with the published results of Li et al.^[1].

-The detail of the Raman experiments is missing from the Experimental section.

Added.

-Figure 1 has some text cut off at the side. Parts b and c also do not make clear the different reactions - could the authors add the weight fractions of the starting materials as well for clarity?

Fixed the cut-off text. Also, the caption has been updated to clarify the origins of the two phases, while the wt% of the precursors has been added to the main text.

-The layout of Table 1 is visually confusing. Would it be possible to have the lattice parameters and goodness of fit details in separate rows to the site details?

We explored many layouts of the Table 1. We found that the one included is the best for containing all the relevant data in a visually appealing and logical format. We have added some additional spacing and bold text to help facilitate ease of reading the table.

-There are a number of formatting inconsistencies throughout the manuscript and SI which need to be addressed (e.g. capitalisation, chemical formulae, introduction of QENS acronym)

We have reviewed the text and standardized the formatting.

Reviewer #2 (Remarks to the Author):

The manuscript “Order-Disorder and Ionic Conductivity in Calcium Nitride-Hydride Ammonia Synthesis Catalysts” presents results on two phases of Ca_2NH , one of which shows reasonably high hydride ionic conductivity. In this respect, the work builds on recent developments in the solid-state hydride conductor literature, particularly with regard to mixed-anionic conductors (oxyhydrides, nitride hydrides) that have attracted recent interest.

The data collection and analysis are thorough and appear to be well documented for reproducibility, though the discussion and presentation are at times confusing. It is not always clear whether the authors have justifications for some of the assumptions they make. In several cases, computational analysis would be an appropriate means to support the authors' claims. Among the comments that caught my attention:

Computational analysis is definitely of future interest but is typically challenging in disordered and short range order systems that are core to this discussion. Hopefully our data can facilitate such data to come forward. One aspect of particular interest is the thermodynamics of vacancy creation and the effects of the presence of the imide species which could be addressed.

On pg 6: “Since, the nH2 site is associated with intrinsic anti-Frenkel defects, nH2 is equal to the vacancy concentration in the octahedral position ($n_{\text{v},\text{H1}}$).” This seems to assume two things: one, that there is an exact stoichiometric concentration of hydrogen in the system, and two, that hydrogen can reside solely on the octahedral or tetrahedral sites (not, for instance, on nominal N sites). Can these assumptions be justified? If there are lots of H atoms on

tetrahedral sites, it seems suspicious that there apparently aren't many H1-H2 jumps contributing to the measured conductivity.

This has been clarified, "Since, the n_{H2} site is associated with intrinsic anti-Frenkel defects, n_{H2} should reflect the vacancy concentration in the octahedral position ($n_{v,H1}$) for this β Ca_2NH structure which refines as overall stoichiometric"

"Full refinement yielded an occupancy of the D1 octahedral site of 0.822(11) as opposed to the value of 0.75 reported by Brice et al., with the remainder being located in a tetrahedral site (18.9%, as opposed to 25% previously published) giving a stoichiometric compound ($\text{Ca}_2\text{NH}_{0.997(34)}$)."

Of α - Ca_2NH , on pg 6: "there are no vacancies in the octahedral sites likely as a result of the entropic contribution of the imide species." I assume that, in a perfect structure, all H in α - Ca_2NH would reside on octahedral sites. If that is the case, full octahedral occupation ("no vacancies", as the authors say), plus protonic interstitials (the imides), would add up to significant excess hydrogen. Does α - Ca_2NH contain a significant H excess?

Yes, the refinement gives a stoichiometry of $\text{Ca}_2\text{N}_{0.8}\text{H}_{0.8}(\text{NH})_{0.4}$. This means 20% of the octahedral positions are occupied by imide species (NH^{2-}) which gives a hydrogen to calcium ratio of 1.2:2 compared with 1:2 for β - Ca_2NH .

On pg 7, comparing Ca_2NH to BaH_2 : "the difference arising from the significantly higher charge carrier concentrations in the BaH_2 due to the presence of a concerted migration mechanism." Previous work, including reports from some of the same authors, has shown clearly that BaH_2 has a vacancy-mediated conduction pathway, which seems analogous to that reported here for Ca_2NH . Thus, invoking a different "migration mechanism" to explain the difference in conductivities does not make sense to me.

There are certainly vacancies in BaH_2 but that does not mean that there is not concerted migration. Indeed inter-site distances are too small to allow unconstrained occupation. We have recently published a paper that takes a close look at the conduction mechanism of BaH_2 and concluded that is a concerted migration mechanism[2].

The caption of Fig. S4 states: "It is expected from thermal vibrations that peaks would broaden with increasing temperature. In this case, we see the opposite trend." Can the authors explain this discrepancy?

In the main text, this figure is referred to as characteristic for systems containing secondary anion species with citations provided. We have added a sentence to the caption to clarify the significance. "It is expected from thermal vibrations that peaks would broaden with increasing temperature. In this case, we see the opposite trend as a result of the increase of the secondary species in the lattice."

Additional information regarding the stability of these structures is also requested. Specifically, in light of the observation of SrO in several diffraction plots, how stable are they in ambient conditions? This information is important for further experimental development. And which of the two phases in question (alpha or beta) is thermodynamically preferred?

Added language to discuss stability and handling of materials to the experimental section. The thermodynamic preference is not clear and should be explored in future work, specifically computational. However, we note here that both the β - and α -phases reported herein are likely metastable given their dependence on synthesis routine.

The presentation of the structures is also confusing. In place of, or in addition to, the text description of the phases, I recommend modifying Fig S1 to show explicitly the crystal structures under consideration and label them as α -Ca₂NH and β -Ca₂NH. Including the R-3m phase doesn't seem necessary, since it doesn't participate in the authors' discussion, but if it is discussed, it should be assigned its own Greek letter for clarity. Also, please indicate the H1 and H2 sites, which are invoked in discussions of H transport.

The R-3m structure is necessary to show how the space group assignment affects the dimensionality of the diffusive process, with the R-3m assignment resulting in a 2D diffusion and the Fd-3m being 3D. We have added detail to figure S1 to make the diffusion pathways more clear, and a third figure to show the position of the tetrahedral position.

Addressing these concerns is necessary for me to recommend this work as a contribution to Nature Communications.

Reviewer #3 (Remarks to the Author):

This paper reports on Order-Disorder and Ionic Conductivity in Calcium Nitride-Hydride Ammonia Synthesis Catalysts. The paper is well written and easy to follow, and the subject is relevant. The approach used is valid and the conclusions being drawn from results are correct. The manuscript can be accepted after the following minor revisions:

1) The notation of Space Group is not correct: please change Fd-3m and R-3m using italic for Lattice and mirror plane

Changed.

2) Wyckoff letter (i.e. 96g, 48f) must be italic (96g, 48f).

Changed

3) The authors should always specify:

- Radiation type and wavelength.
- Data collection conditions: 2θ range, step size, time per step;

Added a citation to Polaris diffractometer that covers all these parameters. For the XRD patterns, specified this information. This information has been added to the XRD pattern in the SI.

- The software used for the Rietveld refinement.

Added

- The structural models used for the full-profile fit refinement (giving the sources: literature reference, database or phase code);

Added

- If a correction for preferred orientation effects (e.g., March-Dollase function, or spherical harmonics) has been applied.

No, preferred orientation models were used.

- No. of variables,
- No. of observations
- No. of reflections
- R_p (%), R_{wp} (%) and RF^{*2} (%)
- residual electron density (min/max $e\text{\AA}^{-3}$)

I have added these additional statistics.

4) Page 4, please report the Standard errors of unit cell parameters in parentheses (“The relative lattice parameters for two polymorphs are 10.30 Å vs 10.23 Å for the α - and β -phases respectively....”).

Added

5) Table 1: please comment the very high standard errors of Uiso for D2 (3.17(372) and 5.39(176)). These value are not acceptable. The refinement of these parameters must be revised.

We revisited the errors associated with these sites. For the proton site, due to the low occupancy, and highly disordered nature the Uiso was fixed for this site. For the secondary hydride site, the large error reported in the original manuscript was a typo. The table has been updated with the correct value.

Reviewer #4 (Remarks to the Author):

This paper reports order-disorder and ionic conductivity in Ca-N-H ammonia catalysis. Distinct hydride conductivity difference between the ordered (beta-phase) and disordered phase (alpha phase) in Ca₂NH was experimentally clarified and the microscopic mechanism on high conductivity was well explained on the data evaluated from QENS measurement. However, the claim on the catalyst for NH₃ is too speculative at the present stage: First, the catalytic activity of transition-metal unloaded Ca₂NH is not so high. Thus, Ca₂NH is not the main catalyst but may be regarded as co-catalyst. Second, the activation energy of NH₃ synthesis is much higher than that of hydride ion conduction as reported in Adv. Energ. Mat. 11, 2003723(2021). This fact means the rate

determining step of the NH₃ synthesis is not ionic diffusion. It is likely the formation of low work function electron at the site of hydride vacancy plays a crucial for N₂ activation through electron donation to loaded transition metal. So, hydride vacancy works well for this purpose. This model is supported by a striking difference in the catalytic activity between Ca₂NH and CaNH (Chem.Sci. 7, 4036(2016)).

The authors should confine the main claim which can be supported by the solid experimental results.

The experiment results have been put into context through the addition of a discussion on ammonia synthesis rates for CaH₂ and Ca₂NH loaded with Ru. We agree that these materials act as co-catalysts in the reported studies where they support a high surface activity metal catalyst.

Minor comments:

1. No QENS data were shown and experimental description is too brief.

QENS data are shown in Figure S10. We have added details about the QENS experiment to the main text.

2. There are several types such as Table S1 (3rd column, Ca₂Ne-+½N₂ →Ca₂NH, N₂=>H₂) and incomplete sentences in the text.

We have corrected this specific mistake and go through the text and corrected other grammatical and formatting issues.

References:

- [1] W. Li, G. Wu, Z. Xiong, Y. P. Feng, and P. Chen, “Li + ionic conductivities and diffusion mechanisms in Li-based imides and lithium amide,” *Phys. Chem. Chem. Phys.*, vol. 14, pp. 1596–1606, 2012, doi: 10.1039/c2cp23636b.
- [2] G. J. Irvine, F. Demmel, H. Y. Playford, G. Carins, M. O. Jones, and J. T. S. Irvine, “Geometric Frustration and Concerted Migration in the Superionic Conductor Barium Hydride,” *Chem. Mater.*, 2022, doi: 10.1021/acs.chemmater.2c01995.

REVIEWERS' COMMENTS

Reviewer #1 (Remarks to the Author):

The response to the reviews adequately addresses my feedback.

Reviewer #2 (Remarks to the Author):

The authors have adequately addressed my comments in revising their manuscript. I have no further comments prohibiting its publication.

Reviewer #3 (Remarks to the Author):

The manuscript was modified and is suitable for publication.

Reviewer #4 (Remarks to the Author):

The drawbacks I raised were amended. I think this paper reaches the level acceptable by Nat.Com.

Response to Reviewers:

We want to again thank the reviewers for their time and helpful feedback. We also want to thank for the timely feedback to our reply to their comments.

RESPONSE TO REVIEWERS' COMMENTS

Reviewer #1 (Remarks to the Author):

The response to the reviews adequately addresses my feedback.

Reviewer #2 (Remarks to the Author):

The authors have adequately addressed my comments in revising their manuscript. I have no further comments prohibiting its publication.

Reviewer #3 (Remarks to the Author):

The manuscript was modified and is suitable for publication.

Reviewer #4 (Remarks to the Author):

The drawbacks I raised were amended. I think this paper reaches the level acceptable by Nat.Com.